# Exploring contextual effects of post-migration housing environment on mental health of asylum seekers and refugees: A cross-sectional, population-based, multi-level analysis in a German federal state

Amir Mohsenpour[1,2,3]*, Louise Biddle[1,2], Kayvan Bozorgmehr[1,2]

1 Department of Population Medicine and Health Services Research, School of Public Health, Bielefeld University, Bielefeld, Germany, 2 Section for Health Equity Studies and Migration, Department of General Practice and Health Services Research, Heidelberg University Hospital, Heidelberg, Germany, 3 Department for Psychiatry, Psychotherapy and Psychosomatic Medicine, Vitos Kurhessen, Kassel, Germany

* amir.mohsenpour@uni-bielefeld.de

**Data Availability Statement:** This research is based on sensitive human research participant data

## Abstract

Asylum seekers and refugees (ASR) in Germany are dispersed quasi-randomly to state-provided, collective accommodation centres. We aimed to analyse contextual effects of post-migration housing environment on their mental health. We drew a balanced random sample of 54 from 1 938 accommodation centres with 70 634 ASR in Germany's 3rd largest federal state. Individual-level data on depression and anxiety as well as sociodemographic- and asylum-related covariates, were collected and linked to contextual geo-referenced data on housing environment ('Small-area Housing Environment Deterioration' index, number of residents, remoteness, urbanity, and German Index of Multiple Deprivation). We fitted two-level random-intercept models to exploratively estimate adjusted contextual effects. Of 411 surveyed participants, 45.53% and 44.83%, respectively, reported symptoms of depression or anxiety. 52.8% lived in centres with highest deterioration, 46.2% in centres with > = 50 residents, 76.9% in urban, and 56% in deprived districts. 7.4% of centres were remote. We found statistically significant clustering in reporting anxiety on the level of accommodation centres. The model resulted in an intraclass correlation of 0.16 which translated into a median odds ratio of 2.10 for the accommodation-level effects. No significant clustering was found for symptoms of depression. The highest degree of deterioration, large accommodation size, remoteness, and district urbanity showed higher, but statistically not significant, odds for reporting anxiety or depression. District deprivation demonstrated higher odds for anxiety and lower odds for depression yet remained statistically insignificant for both. Evidence for contextual effects of housing environment on mental health of ASR could not be established but residual confounding by length of stay in the accommodation centre cannot be ruled out. Confirmatory analyses with prior power calculations are needed to complement these exploratory estimates.

containing potentially identifying information. Participants have agreed that their data may be used for non-profit research purposes by academic institutions. Data and survey instruments are available upon request through Respond. AMED@med.uni-heidelberg.de.

**Funding:** The primary data collection was funded by the German Federal Ministry for Education and Research (BMBF) in the scope of the project RESPOND (Grant Number: 01GY1611; Principal applicant: KB). The analysis was realised with funding of the German Science Foundation (DFG) in the scope of the NEXUS project (FOR 2928 / GZ: BO 5233/1-1; Principal applicant: KB). AM acknowledges financial support by the Else Kröner-Fresenius-Stiftung (2017_Promotionskolleg.08) within the Heidelberg Graduate School of Global Health. The funders had no influence on study design, data collection and analysis, or decision to publish, or preparation of the manuscript.

**Competing interests:** The authors have declared that no competing interests exist.

## Introduction

Undergoing disruptive situations in their home country, during flight or in arrival countries, asylum seekers and refugees (ASR) experience a high burden of disease [1–3]. This is particularly reflected in high prevalence rates of psychological distress, e. g. depression, generalized anxiety disorder or post-traumatic stress, both globally [4, 5] and in Germany [1, 4, 6, 7].

Mental health research has previously focused on pre- and peri-migratory risk factors when studying refugee health [8], but focus has been slowly shifting towards post-migration stressors in the country of reception, e. g. unemployment, loneliness or factors relating to the asylum process [6, 9–11].

In addition to these individual-level factors, several contextual determinants of mental health have been discussed. These range from housing type and condition [9, 12], residential instability [13] or economic opportunity [9] to restrictive (non-)health migration policies [14, 15]. Their importance has been particularly aggravated by the SARS-CoV-2 pandemic where ASR have been experiencing a high cumulative incidence risk in case of an outbreak within their accommodation [16].

Furthermore, the German asylum system assigns newly arriving ASR to a place of residence based on administrative quota in a quasi-random manner, and free movement to other residential areas is restricted until the application is closed. Such assignment to neighbourhoods is not only disempowering and limits individual autonomy [17], but has resulted in higher numbers of vulnerable asylum seekers (i.e. minors, female, or elderly) living in districts characterised by high socioeconomic deprivation [18]. Such disadvantageous contextual conditions may exacerbate pre-existing socioeconomic inequalities, aggravate (perceived) downward social mobility, and deteriorate the already high burden of mental illness [19].

However, research on small-area housing environment and its potential health effects is still scarce, and often limited by crude or non-standardised measurements on housing characteristics, high-levels of aggregation of contextual factors, and risk of compositional bias due to selective migration into housing environments. For example, housing measures have been criticized for being not rigorous, very diverse, and mainly based on respondent self-reports with researcher observations being the exception.

[12] Others report on crude measures such as "private/shared" accommodation [9], while more sophisticated analyses of housing environments predominantly include populations with completed asylum claims [20] and hence suffer from the risk of selective migration into residential areas.

Against these backdrops, we aimed to exploratively assess contextual effects of housing environment on mental health of ASR, while using reliable and valid data on the post-migration contextual environment at small-area level and minimising the risk of compositional bias.

## Methods

### Setting, sampling and recruitment

We collected data in 54 collective refugee accommodation centres, sampled from a total of 1 938 centres with 70 634 ASR across all districts of Germany's 3rd largest federal state of Baden-Wuerttemberg (2017/2018). With 10.8 million inhabitants and 44 districts, it receives about 13% of all incoming ASR to Germany based on a quota system considering state-level tax income and population size [21]. After a stay in reception centres for up to 18 months, ASR are quasi-randomly dispersed to district-level collective accommodation centres determined by a quota based on district-level population size. The numbers of ASR assigned to each district are proportional to the size of the district population relative to the overall state

population. As there are no mandatory standards for accommodation centres, location and quality of centres vary strongly within and between districts [22].

Participants were recruited using a complex random sampling design balanced for (1) the number of resident ASR within centres and (2) their total number in the region. Questionnaires for data collection were developed in English and German, and made available in seven additional languages (Albanian, Arabic, Farsi, French, Russian, Serbian, Turkish) based on the prevalent languages in the ASR population. Detailed information on sampling and data collection have been reported previously [2, 23].

Asylum seekers were involved in the design of the questionnaire by means of pre-testing selected instruments and incorporating their feedback on comprehensibility and linguistic diversity [24].

Furthermore, the design was informed by preceding extensive qualitative studies on living situations in accommodation centres [25–27] which gave voice to asylum seekers regarding their social determinants of health and the related challenges in camps and shared accommodation facilities. The insights from these emic perspectives informed the focus on accommodation centres.

## Ethical considerations

Primary data has been collected as part of the RESPOND project at Heidelberg University Hospital, Germany (https://respond-study.org/) and oral informed consent was obtained from all participants after oral, written and audio-based information in nine languages deployed by a multi-lingual field team. Ethical clearance was received by the ethics committee of the Medical Faculty of Heidelberg on October 12, 2017 (S- 516/2017 [2]).

## Individual-level health and socio-demographic variables

Individual-level variables were captured part of a health monitoring survey using a paper-based questionnaire in nine languages [2, 23]. Mental health was captured by 2-item screeners for symptoms of depression (Public Health Questionnaire-2, PHQ2 [28]) and generalized anxiety (Generalized Anxiety Disorder-2, GAD2 [29]). Both offer a valid ultra-brief tool to identify individuals who may be suffering from either mental disorder.

We further collected data on participants' socio-demographics (age at interview, sex, educational attainment, region of origin, the number of children), physical health (self-reported chronic illness), and factors related to the asylum process (residence status) including the number of transfers between accommodation centres as a proxy of residential instability as these have been reported to be influential for mental health [3, 11, 13]. We used these factors to adjust for potential confounding by individual-level health, socio-demographic characteristics, and aspects related to the asylum process (Table 1).

## Contextual variables on housing environment

We linked individual-level data of ASR with contextual characteristics of their housing environment at level of accommodation centres (via unique identifiers for each centre and participating individual). Centre data (at level of exact streets and houses) was linked to neighbourhood characteristics via Google Maps, and to district-level characteristics (nomenclature des unités territoriales statistiques (NUTS) level 3) via secondary data-sources and official statistics (Table 1).

Data at level of accommodation centres comprised the accommodation size, i.e. number of residents, and the stage of decay of the built environment. The size cut-off at 50 residents per accommodation centre was based on thorough discussions in the team and the descriptive

**Table 1. Data dictionary explaining outcomes, exposures, and covariates.**

|  | Definition |
|---|---|
| **Outcomes of interest** |  |
| Generalized Anxiety | 2-item short version (GAD-2) of 7-item Generalized Anxiety scale. Outcome coded binary at > = 3 points. |
| Depression | 2-item short version (PHQ-2) of 9-item Patient Health Questionnaire depression module. Outcome coded binary at > = 3 points. |
| **Exposures of interest** |  |
| SHED index | Small-area Housing Environment Deterioration (SHED) index, covering five dimensions of physical environment and their degree of deterioration, i.e. (1) conditions of windows/glass and (2) walls/roof, (3) garbage accumulation inside/outside the house, (4) graffiti inside/outside the house and (5) outside spaces and complemented by a global rating of the overall living environment as a sixth item. SHED coded as quartiles: Q1 (lowest level of deterioration) to Q4 (highest level of deterioration). |
| No. of residents | Number of residents per collective accommodation centre. Dichotomized at > = 50 residents. |
| Remoteness | Remoteness index calculated based on Google maps, composed of travel distance to three locations, (1) general medical facilities, (2) grocery shopping facilities and (3) town hall as a central location. Measured at Monday mornings at 9am local time, coded binary at > = 2 locations needing >20minutes (both on foot and by public transportation). |
| Urbanity | District urbanity, as classified by the German Federal Institute for Research on Building, Urban Affairs and Spatial Development (BBSR). |
| GIMD-10 | District deprivation, as classified by the German Index of Multiple Deprivation, Version 10 (GIMD-10). Dichotomized (at median cut-off) in high/low deprivation |
| **Covariates for adjustment** |  |
| Age at interview | Age in full years, calculated based on self-reported date of birth. |
| Sex | Self-reported sex (female/male). |
| Self-reported chronic illness | Self-reported longstanding illness (yes/no). |
| Region of origin | Region of origin based on self-reported country of origin (Europe, Asia, Africa, other) |
| Educational score | Composite educational score based on self-reported highest educational attainment and highest professional education (1–6; low to high). |
| Number of children | Self-reported number of children (0 or > = 1) |
| Asylum residence status | Self-reported asylum residence status (rejected/only tolerated or to be decided/granted) |
| Accommodation transfers | Self-reported number of accommodation transfers (0–1 or > = 2) |

GAD-2: Kroenke K, Spitzer RL, Williams JB, Löwe B. An ultra-brief screening scale for anxiety and depression: the PHQ—4. Psychosomatics. 2009; 50(6):613–21.

PHQ-2: Kroenke K, Spitzer RL, Williams JBW. The Patient Health Questionnaire-2: Validity of a two-item depression screener. Med Care. 2003;41(11):1284–92.

analyses of the data. The stage of decay was quantified by the Small-area Housing Environment Deterioration index (SHED). SHED was rated by the field teams and consists of six dimensions assessing (1) conditions of windows/glass and (2) walls/roof, (3) garbage accumulation and (4) graffiti inside and outside the house and (5) the condition of outside spaces including gardens, complemented by (6) a global rating of the overall living environment. SHED's theoretical framework, its construction, piloting, and validation, including proven high intra- and interrater reliability, have been reported elsewhere in detail [30]. The composite SHED index score is calculated based on individual item's z-transformation (to standardize item scores across different scales) and normalization (to scale all item scores in the range of 0–1). We then split the index score into four relative quartiles of distribution, ranging from Q1 (lowest) to Q4 (highest level of deterioration).

Neighbourhood characteristics for each accommodation centre included a remoteness index. Using Google Maps and each centre's exact address, we calculated the travel distance to three locations of essential services (medical services, grocery shopping, town hall), both on foot and using public transportation, for Monday mornings at 9am. After, we dichotomized the index at $\geq 2$ locations needing $> 20$ minutes both on food and by public transportation (if available). 20 minutes is the time period which the German Association of Statutory Health Insurance Physicians (*Kassenärztliche Vereinigung*) considers as acceptable and uses for their own ambulatory health care planning [31]. Accommodations were then linked with district-level factors comprising urbanity, as assessed by the German Federal Institute for Research on Building, Urban Affairs and Spatial Developments (BBSR) defining an urban district as one with a population density over 150 inhabitants per $km^2$ [32], as well as district deprivation, as assessed by the German Index of Multiple Deprivation, Version 2010 [33] calibrated to the distribution of the GIMD score within the state of Baden-Württemberg. Centres were then classified as located in 'high-/low- deprivation districts' based on the median GIMD-10 score across all 44 districts.

## Conceptual model

We conceptualised the relationship between exposures, outcomes, and co-variables vis à vis the dispersion process of being assigned to residential areas in a causal diagram guiding our analysis. The dispersion functions as a quasi-random allocation of asylum seekers to district-level collective accommodation centres. See Fig 1.

## Statistical analyses

We performed a descriptive analysis of absolute and relative frequencies for each contextual exposure variable at the level of accommodation centres. We further quantified the individual-

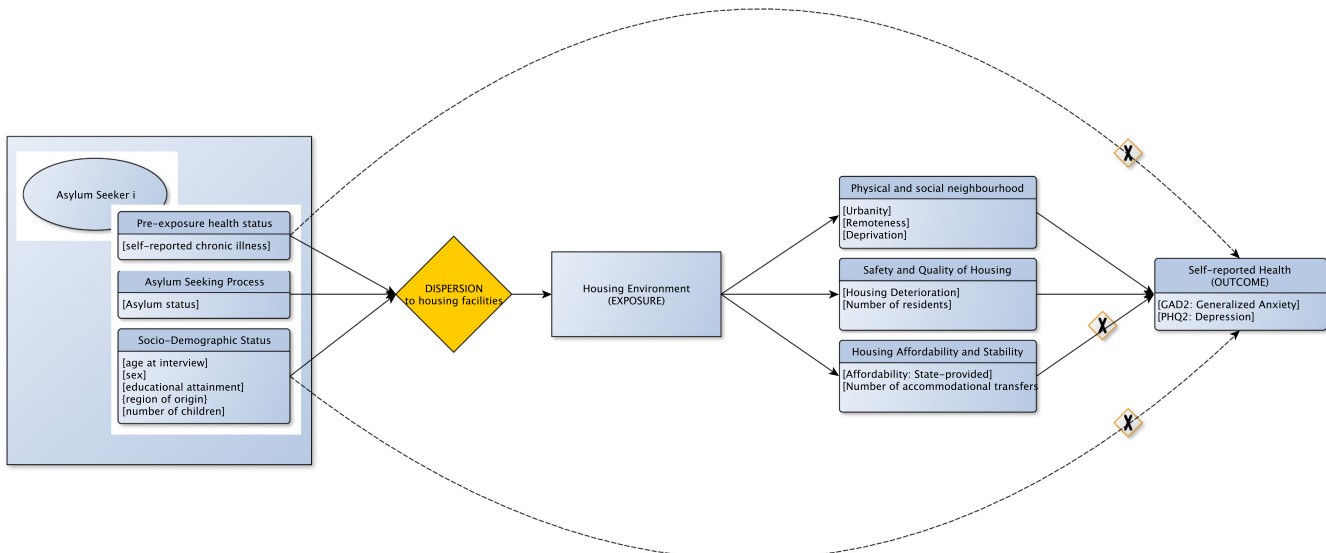

**Fig 1. Causal diagram on housing environment and health.** Caption: Causal diagram on housing environment and health guiding our analyses along the dispersion process of being assigned to residential areas with time flowing from left to right. Effects of housing on self-reported health are mediated via (1) physical and social neighbourhood, (2) safety and quality of housing and (3) housing affordability and stability. Focussing on the contextual pathways, effects through number (3) are blocked as (a) state-provided accommodations nullify questions of affordability and (b) our models are adjusted for the number of accommodational transfers conceptualized as housing stability. Other potential confounders (pre-exposure health status as well as sociodemographic factors) are adjusted for in the models. Time flows from left to right. Variables for each domain have been put into square brackets.

level distribution of the outcome variables of interest stratified by all contextual measures and potentially confounding covariates.

Two-level random-intercept logistic regression models were fitted for each outcome variable of interest (PHQ2, GAD2) calculating odds ratios (OR) and 95% confidence intervals (95% CI) adjusting for above mentioned individual-level covariates and the clustering of participant-level observations within each accommodation centre.

For this, we utilized a stepwise approach starting with fitting a null model displaying accommodation-level variability in outcome variables (M0) and assessing statistically significant clustering in such two-level random-intercept models compared to single-level null-models. Next, we fitted a two-level model for each outcome and each exposure individually, both unadjusted, i.e. without individual-level variables (M1a-M1e), and adjusted, i.e. including individual-level variables (M2a-M2e), before fitting a multiple regression model for all post-migration contextual housing variables together (unadjusted for individual-level variables) (M3). Finally, we fitted a last model including all post-migration contextual housing variables while adjusting for all above mentioned individual-level variables as potential confounders (M4). We refrained from reporting effect estimates for covariates conceived as potential confounders in our reporting to prevent 'Table 2 fallacy' [34, 35].

To quantify accommodation-level differences in the outcomes of interest, we calculated both the accommodation-level and total variance as well as the intraclass correlation (ICC) as their quotient. Given difficulty in interpretation of ICC in logistic regression as individual-

**Table 2. Describing distribution of contextual variables by accommodation centres and residents.**

| | Collective accommodation centres | | | Residents | | |
|---|---|---|---|---|---|---|
| | Freq. | Col % | Cum % | Freq. | Col % | Cum % |
| **SHED index (quartiles)** | | | | | | |
| Q1 (lowest deterioration) | 19 | 35.19 | 35.19 | 104 | 25.30 | 25.30 |
| Q2 | 10 | 18.52 | 53.70 | 90 | 21.90 | 47.20 |
| Q3 | 17 | 31.48 | 85.19 | 153 | 37.23 | 84.43 |
| Q4 (highest deterioration) | 8 | 14.81 | 100.0 | 64 | 15.57 | 100.00 |
| **Total** | 54 | 100.0 | | 411 | 100.0 | |
| **Accommodation size** | | | | | | |
| <50 residents | 43 | 79.6 | 79.6 | 221 | 53.8 | 53.8 |
| > = 50 residents | 11 | 20.4 | 100.0 | 190 | 46.2 | 100.0 |
| **Total** | 54 | 100.0 | | 411 | 100.0 | |
| **Remoteness** | | | | | | |
| not remote | 50 | 92.6 | 92.6 | 398 | 96.8 | 96.8 |
| remote | 4 | 7.4 | 100.0 | 13 | 3.2 | 100.0 |
| **Total** | 54 | 100.0 | | 411 | 100.0 | |
| **Urbanity** | | | | | | |
| rural | 13 | 24.1 | 24.1 | 95 | 23.1 | 23.1 |
| urban | 41 | 75.9 | 100.0 | 316 | 76.9 | 100.0 |
| **Total** | 54 | 100.0 | | 411 | 100.0 | |
| **GIMD-10** | | | | | | |
| low deprivation | 24 | 44.4 | 44.4 | 181 | 44.0 | 44.0 |
| high deprivation | 30 | 55.6 | 100.0 | 230 | 56.0 | 100.0 |
| **Total** | 54 | 100.0 | | 411 | 100.0 | |

SHED: Small-area Housing Environment Deterioration

GIMD-10: German Index of Multiple Deprivation, Version 10

level and contextual-level variance are not directly comparable [36, 37], we further calculated median odds ratios (MOR) of the distribution of odds ratios calculated for each pair of participants with similar individual-level covariates across clusters to better understand differences in outcomes based on accommodation centres.

For comparing and assessing model fit, we calculated Akaike information criterion (AIC), Bayesian information criterion (BIC) and model significance based on Wald-chi2. To assess potential multicollinearity, we further computed the variance inflation factor (VIF) for each variable and reported the range of VIFs for each model.

For visualization purposes, we plotted the predicted outcomes of interest against each accommodation centre, based on the null, exposure adjusted and fully adjusted models.

For all statistical analyses, we utilized an approach based on listwise deletion of missing data, determined the level of statistical significance at a p-value of 0.05 and conducted all computation in Stata SE V16 [38].

## Results

### Post-migration contextual housing environment

Of all randomly selected accommodation centres, 46.29% presented with high (Q3) or highest levels (Q4) of deterioration on the SHED index (Table 2). In terms of residents, this translates to 52.8% of study participants living in such deteriorated housing. Most of the accommodation centres (79.6%) had less than 50 ASR as residents, while the average number of residents was ca. 34 (SD = 34, min = 4 to max = 158). At the same time, the 20.4% accommodation centres with 50 or more residents accounted for 46.2% of all ASR in the sample. While most ASR (76.9%) were living in urban districts, 56% were living in districts characterised by high levels of deprivation. A total of 7.4% of accommodation centres, accounting for 3.2% of ASR, were remotely located.

### Distribution of health outcomes by individual-level characteristics

Of all surveyed ASR, 45.53% reported symptoms of depression and 44.83% symptoms of generalized anxiety. The prevalence of symptoms of depression or anxiety were comparable across the strata of age, sex, self-reported longstanding illness, region of origin, educational attainment, number of children, or residence status (Table 3). A large majority (more than 80%) of those with symptoms of depression or anxiety reported two or more accommodational transfers since arrival in Germany (Table 3) with about 50% awaiting results of their asylum application.

### Distribution of health outcomes by accommodation-level characteristics

Exploring accommodation-level characteristics, participants reporting symptoms of depression or generalized anxiety shared another similar pattern: Equal numbers of residents lived in accommodation centres with 50 or more residents (50.6% depression; 54.5% generalized anxiety) and a majority in districts classified as urban (75.9%; 80.8%) and as deprived (55.1%; 60.3%). 5.1% and 5.8% of symptom-positive participants, respectively, lived in accommodation centres assessed as remote. In terms of the SHED index, the largest group of symptom-positive participants lived in housing of the third highest level of deterioration (39.9%; 30.1%). For further details, please consult Table 3.

### Generalized linear models

Fitting an empty two-level random-intercepts model, we found statistically significant clustering in reporting symptoms of generalized anxiety on the level of accommodation centres

**Table 3. Absolute and relative frequencies of symptoms of depression and anxiety by individual- and contextual-level variables.**

| | Symptoms of Depression (PHQ2) | | | | | | | | | Symptoms of Generalized Anxiety (GAD2) | | | | | | | | |
| --- | --- | --- | --- | --- | --- | --- | --- | --- | --- | --- | --- | --- | --- | --- | --- | --- | --- | --- |
| | No | | | Yes | | | Total | | | No | | | Yes | | | Total | | |
| | Freq. | Col % | Cum % | Freq. | Col % | Cum % | No. | Col % | Cum % | Freq. | Col % | Cum % | Freq. | Col % | Cum % | Freq. | Col % | Cum % |
| **Age at interview** | | | | | | | | | | | | | | | | | | |
| 18–25 | 58 | 33.9 | 33.9 | 45 | 30.8 | 30.8 | 103 | 32.5 | 32.5 | 56 | 32.9 | 32.9 | 48 | 32.7 | 32.7 | 104 | 32.8 | 32.8 |
| 26–30 | 33 | 19.3 | 53.2 | 22 | 15.1 | 45.9 | 55 | 17.4 | 49.8 | 35 | 20.6 | 53.5 | 21 | 14.3 | 46.9 | 56 | 17.7 | 50.5 |
| 31–35 | 28 | 16.4 | 69.6 | 24 | 16.4 | 62.3 | 52 | 16.4 | 66.2 | 27 | 15.9 | 69.4 | 26 | 17.7 | 64.6 | 53 | 16.7 | 67.2 |
| 36–40 | 24 | 14.0 | 83.6 | 23 | 15.8 | 78.1 | 47 | 14.8 | 81.1 | 23 | 13.5 | 82.9 | 21 | 14.3 | 78.9 | 44 | 13.9 | 81.1 |
| 41+ | 28 | 16.4 | 100.0 | 32 | 21.9 | 100.0 | 60 | 18.9 | 100.0 | 29 | 17.1 | 100.0 | 31 | 21.1 | 100.0 | 60 | 18.9 | 100.0 |
| **Total** | 171 | 100.0 | | 146 | 100.0 | | 317 | 100.0 | | 170 | 100.0 | | 147 | 100.0 | | 317 | 100.0 | |
| **Sex** | | | | | | | | | | | | | | | | | | |
| male | 120 | 67.4 | 67.4 | 100 | 68.0 | 68.0 | 220 | 67.7 | 67.7 | 127 | 70.6 | 70.6 | 96 | 65.3 | 65.3 | 223 | 68.2 | 68.2 |
| female | 58 | 32.6 | 100.0 | 47 | 32.0 | 100.0 | 105 | 32.3 | 100.0 | 53 | 29.4 | 100.0 | 51 | 34.7 | 100.0 | 104 | 31.8 | 100.0 |
| **Total** | 178 | 100.0 | | 147 | 100.0 | | 325 | 100.0 | | 180 | 100.0 | | 147 | 100.0 | | 327 | 100.0 | |
| **Longstanding illness** | | | | | | | | | | | | | | | | | | |
| no | 118 | 67.4 | 67.4 | 70 | 48.3 | 48.3 | 188 | 58.8 | 58.8 | 122 | 69.7 | 69.7 | 66 | 45.5 | 45.5 | 188 | 58.8 | 58.8 |
| yes | 57 | 32.6 | 100.0 | 75 | 51.7 | 100.0 | 132 | 41.2 | 100.0 | 53 | 30.3 | 100.0 | 79 | 54.5 | 100.0 | 132 | 41.2 | 100.0 |
| **Total** | 175 | 100.0 | | 145 | 100.0 | | 320 | 100.0 | | 175 | 100.0 | | 145 | 100.0 | | 320 | 100.0 | |
| **Region of origin** | | | | | | | | | | | | | | | | | | |
| Europe | 9 | 5.1 | 5.1 | 6 | 4.0 | 4.0 | 15 | 4.6 | 4.6 | 10 | 5.6 | 5.6 | 4 | 2.7 | 2.7 | 14 | 4.3 | 4.3 |
| Asia | 113 | 63.5 | 68.5 | 96 | 64.4 | 68.5 | 209 | 63.9 | 68.5 | 109 | 61.2 | 66.9 | 102 | 68.5 | 71.1 | 211 | 64.5 | 68.8 |
| Africa | 30 | 16.9 | 85.4 | 34 | 22.8 | 91.3 | 64 | 19.6 | 88.1 | 33 | 18.5 | 85.4 | 31 | 20.8 | 91.9 | 64 | 19.6 | 88.4 |
| other | 26 | 14.6 | 100.0 | 13 | 8.7 | 100.0 | 39 | 11.9 | 100.0 | 26 | 14.6 | 100.0 | 12 | 8.1 | 100.0 | 38 | 11.6 | 100.0 |
| **Total** | 178 | 100.0 | | 149 | 100.0 | | 327 | 100.0 | | 178 | 100.0 | | 149 | 100.0 | | 327 | 100.0 | |
| **Educational score** | | | | | | | | | | | | | | | | | | |
| low | 56 | 40.0 | 40.0 | 31 | 26.3 | 26.3 | 87 | 33.7 | 33.7 | 49 | 34.3 | 34.3 | 39 | 33.9 | 33.9 | 88 | 34.1 | 34.1 |
| medium | 55 | 39.3 | 79.3 | 60 | 50.8 | 77.1 | 115 | 44.6 | 78.3 | 60 | 42.0 | 76.2 | 54 | 47.0 | 80.9 | 114 | 44.2 | 78.3 |
| high | 29 | 20.7 | 100.0 | 27 | 22.9 | 100.0 | 56 | 21.7 | 100.0 | 34 | 23.8 | 100.0 | 22 | 19.1 | 100.0 | 56 | 21.7 | 100.0 |
| **Total** | 140 | 100.0 | | 118 | 100.0 | | 258 | 100.0 | | 143 | 100.0 | | 115 | 100.0 | | 258 | 100.0 | |
| **Children** | | | | | | | | | | | | | | | | | | |
| no children | 77 | 40.7 | 40.7 | 62 | 39.2 | 39.2 | 139 | 40.1 | 40.1 | 81 | 42.2 | 42.2 | 61 | 39.1 | 39.1 | 142 | 40.8 | 40.8 |
| > = 1 children | 112 | 59.3 | 100.0 | 96 | 60.8 | 100.0 | 208 | 59.9 | 100.0 | 111 | 57.8 | 100.0 | 95 | 60.9 | 100.0 | 206 | 59.2 | 100.0 |
| **Total** | 189 | 100.0 | | 158 | 100.0 | | 347 | 100.0 | | 192 | 100.0 | | 156 | 100.0 | | 348 | 100.0 | |
| **Residence status** | | | | | | | | | | | | | | | | | | |
| Asylum process ongoing | 87 | 54.4 | 54.4 | 69 | 50.0 | 50.0 | 156 | 52.3 | 52.3 | 92 | 55.8 | 55.8 | 68 | 49.6 | 49.6 | 160 | 53.0 | 53.0 |
| Asylum status granted | 39 | 24.4 | 78.8 | 34 | 24.6 | 74.6 | 73 | 24.5 | 76.8 | 43 | 26.1 | 81.8 | 30 | 21.9 | 71.5 | 73 | 24.2 | 77.2 |
| Asylum only tolerated | 12 | 7.5 | 86.2 | 18 | 13.0 | 87.7 | 30 | 10.1 | 86.9 | 12 | 7.3 | 89.1 | 17 | 12.4 | 83.9 | 29 | 9.6 | 86.8 |
| Asylum status rejected | 22 | 13.8 | 100.0 | 17 | 12.3 | 100.0 | 39 | 13.1 | 100.0 | 18 | 10.9 | 100.0 | 22 | 16.1 | 100.0 | 40 | 13.2 | 100.0 |
| **Total** | 160 | 100.0 | | 138 | 100.0 | | 298 | 100.0 | | 165 | 100.0 | | 137 | 100.0 | | 302 | 100.0 | |
| **Number of transfers** | | | | | | | | | | | | | | | | | | |
| 0–1 transfer | 35 | 18.5 | 18.5 | 30 | 19.0 | 19.0 | 65 | 18.7 | 18.7 | 38 | 19.8 | 19.8 | 28 | 17.9 | 17.9 | 66 | 19.0 | 19.0 |
| > = 2 transfers | 154 | 81.5 | 100.0 | 128 | 81.0 | 100.0 | 282 | 81.3 | 100.0 | 154 | 80.2 | 100.0 | 128 | 82.1 | 100.0 | 282 | 81.0 | 100.0 |
| **Total** | 189 | 100.0 | | 158 | 100.0 | | 347 | 100.0 | | 192 | 100.0 | | 156 | 100.0 | | 348 | 100.0 | |
| **SHED, relative quartiles** | | | | | | | | | | | | | | | | | | |
| Q1 (least) | 57 | 30.2 | 30.2 | 33 | 20.9 | 20.9 | 90 | 25.9 | 25.9 | 57 | 29.7 | 29.7 | 35 | 22.4 | 22.4 | 92 | 26.4 | 26.4 |
| Q2 | 43 | 22.8 | 52.9 | 35 | 22.2 | 43.0 | 78 | 22.5 | 48.4 | 34 | 17.7 | 47.4 | 43 | 27.6 | 50.0 | 77 | 22.1 | 48.6 |

*(Continued)*

**Table 3.** (Continued)

| | Symptoms of Depression (PHQ2) | | | | | | | | | Symptoms of Generalized Anxiety (GAD2) | | | | | | | | |
|---|---|---|---|---|---|---|---|---|---|---|---|---|---|---|---|---|---|---|
| | No | | | Yes | | | Total | | | No | | | Yes | | | Total | | |
| | Freq. | Col % | Cum % | Freq. | Col % | Cum % | No. | Col % | Cum % | Freq. | Col % | Cum % | Freq. | Col % | Cum % | Freq. | Col % | Cum % |
| Q3 | 65 | 34.4 | 87.3 | 63 | 39.9 | 82.9 | 128 | 36.9 | 85.3 | 79 | 41.1 | 88.5 | 47 | 30.1 | 80.1 | 126 | 36.2 | 84.8 |
| Q4 (most) | 24 | 12.7 | 100.0 | 27 | 17.1 | 100.0 | 51 | 14.7 | 100.0 | 22 | 11.5 | 100.0 | 31 | 19.9 | 100.0 | 53 | 15.2 | 100.0 |
| **Total** | 189 | 100.0 | | 158 | 100.0 | | 347 | 100.0 | | 192 | 100.0 | | 156 | 100.0 | | 348 | 100.0 | |
| **Accommodation size** | | | | | | | | | | | | | | | | | | |
| <50 | 108 | 57.1 | 57.1 | 78 | 49.4 | 49.4 | 186 | 53.6 | 53.6 | 113 | 58.9 | 58.9 | 71 | 45.5 | 45.5 | 184 | 52.9 | 52.9 |
| > = 50 | 81 | 42.9 | 100.0 | 80 | 50.6 | 100.0 | 161 | 46.4 | 100.0 | 79 | 41.1 | 100.0 | 85 | 54.5 | 100.0 | 164 | 47.1 | 100.0 |
| **Total** | 189 | 100.0 | | 158 | 100.0 | | 347 | 100.0 | | 192 | 100.0 | | 156 | 100.0 | | 348 | 100.0 | |
| **Remoteness** | | | | | | | | | | | | | | | | | | |
| not remote | 184 | 97.4 | 97.4 | 150 | 94.9 | 94.9 | 334 | 96.3 | 96.3 | 188 | 97.9 | 97.9 | 147 | 94.2 | 94.2 | 335 | 96.3 | 96.3 |
| remote | 5 | 2.6 | 100.0 | 8 | 5.1 | 100.0 | 13 | 3.7 | 100.0 | 4 | 2.1 | 100.0 | 9 | 5.8 | 100.0 | 13 | 3.7 | 100.0 |
| **Total** | 189 | 100.0 | | 158 | 100.0 | | 347 | 100.0 | | 192 | 100.0 | | 156 | 100.0 | | 348 | 100.0 | |
| **Urbanity** | | | | | | | | | | | | | | | | | | |
| rural | 39 | 20.6 | 20.6 | 38 | 24.1 | 24.1 | 77 | 22.2 | 22.2 | 47 | 24.5 | 24.5 | 30 | 19.2 | 19.2 | 77 | 22.1 | 22.1 |
| urban | 150 | 79.4 | 100.0 | 120 | 75.9 | 100.0 | 270 | 77.8 | 100.0 | 145 | 75.5 | 100.0 | 126 | 80.8 | 100.0 | 271 | 77.9 | 100.0 |
| **Total** | 189 | 100.0 | | 158 | 100.0 | | 347 | 100.0 | | 192 | 100.0 | | 156 | 100.0 | | 348 | 100.0 | |
| **GIMD-10** | | | | | | | | | | | | | | | | | | |
| low deprivation | 87 | 46.0 | 46.0 | 71 | 44.9 | 44.9 | 158 | 45.5 | 45.5 | 95 | 49.5 | 49.5 | 62 | 39.7 | 39.7 | 157 | 45.1 | 45.1 |
| High deprivation | 102 | 54.0 | 100.0 | 87 | 55.1 | 100.0 | 189 | 54.5 | 100.0 | 97 | 50.5 | 100.0 | 94 | 60.3 | 100.0 | 191 | 54.9 | 100.0 |
| **Total** | 189 | 100.0 | | 158 | 100.0 | | 347 | 100.0 | | 192 | 100.0 | | 156 | 100.0 | | 348 | 100.0 | |

SHED: Small-area Housing Environment Deterioration

GIMD-10: German Index of Multiple Deprivation, Version 10

(Likelihood ratio test; p-value <0.001). The model resulted in an intraclass correlation of 0.16 which translated into a MOR of 2.10 for the accommodation-level effects. No significant clustering at accommodation-level was found for symptoms of depression.

In the univariate and multivariate models, higher odds of reporting symptoms of generalized anxiety were observed for participants living in centres assessed as highly deteriorated (fully adjusted model M4: OR 2.22 [95% CI 0.52,9.59]), accommodating > = 50 residents (1.34 [0.59,3.06]), assessed as remote (2.16 [0.32,14.79]) or situated in urban (3.05 [0.98,9.49]) or deprived (1.21 [0.51,2.88]) districts. Statistical significance was observed for none of the above.

For symptoms of depression, the similar was calculated: Higher odds of reporting symptoms were observed for accommodations with highest level of deterioration (1.99 [0.55,7.18]), accommodating > = 50 residents (1.12 [0.56,2.26]), assessed as remote (3.79 [0.62,23.18]) or those situated in urban districts (1.14 [0.46,2.79]). In contrast to symptoms of generalized anxiety, living in a high-deprivation district may be associated with lower odds of reporting symptoms of depression (0.88 [0.41,1.89]). Similar to GAD2 scores, statistical significance was not observed for any of the above.

The stepwise approach in regression modelling allowed for calculation of several criteria of model fit: Both AIC as well as BIC improved with each model and presented their lowest value in M4. There was no evidence for multicollinearity between exposures based on VIFs. Detailed regression results can be found in Table 4. Predicted random-intercepts for each accommodation centre have been ranked and plotted for three models (M0, M3, M4) visualizing the

**Table 4. Two-level random-intercepts logistic regression of generalized anxiety and depression on contextual housing variables.**

| | Generalized Anxiety (GAD2) | | | | | Depression (PHQ2) | | | | |
|---|---|---|---|---|---|---|---|---|---|---|
| | M0 | M1a-M1e | M2a-M2e | M3 | M4 | M0 | M1a-M1e | M2a-M2e | M3 | M4 |
| | Null | Single contextual exposure, unadjusted | Single contextual exposure, adjusted | Multiple contextual exposures, unadjusted | Multiple contextual exposures, adjusted | Null | Single exposure | Single exposure, adjusted | Multi exposures, unadjusted | Multi exposures, adjusted |
| **Fixed-effects: accommodation-level** | | | | | | | | | | |
| SHED score, quartiles (ref. = Q1, lowest deterioration) | | | | | | | | | | |
| Q2 | | 2.01 | 1.56 | 1.65 | 1.24 | | 1.41 | 0.95 | 1.61 | 0.96 |
| | | [0.84,4.81] | [0.54,4.55] | [0.71,3.81] | [0.40,3.87] | | [0.76,2.61] | [0.39,2.34] | [0.80,3.24] | [0.35,2.67] |
| Q3 | | 0.97 | 0.45 | 0.81 | 0.40 | | 1.67 | 1.74 | 1.78 | 1.86 |
| | | [0.45,2.07] | [0.17,1.18] | [0.39,1.68] | [0.14,1.13] | | [0.96,2.91] | [0.79,3.82] | [0.99,3.22] | [0.78,4.43] |
| (highest deterioration) Q4 | | 2.44 | 2.10 | 1.73 | 2.22 | | 1.94 | 1.85 | 1.92 | 1.99 |
| | | [0.90,6.67] | [0.632,6.95] | [0.62,4.80] | [0.52,9.59] | | [0.97,3.90] | [0.67,5.10] | [0.83,4.44] | [0.55,7.18] |
| No. of residents (ref. = <50) | | | | | | | | | | |
| > = 50 | | 1.59 | 1.21 | 1.81 | 1.34 | | 1.37 | 1.33 | 1.35 | 1.12 |
| | | [0.82,3.10] | [0.52,2.83] | [1.00,3.30] | [0.59,3.06] | | [0.89,2.09] | [0.74,2.41] | [0.85,2.16] | [0.56,2.26] |
| Remoteness (ref. = not remote) | | | | | | | | | | |
| remote | | 3.94 | 4.15 | 3.25 | 2.16 | | 1.96 | 2.91 | 2.70 | 3.79 |
| | | [0.86,18.11] | [0.54,32.19] | [0.81,13.07] | [0.32,14.79] | | [0.63,6.12] | [0.51,16.63] | [0.82,8.86] | [0.62,23.18] |
| Urbanity (ref. = rural) | | | | | | | | | | |
| urban | | 1.54 | 2.91 | 1.53 | 3.05 | | 0.82 | 1.10 | 0.70 | 1.14 |
| | | [0.69,3.46] | [0.928,9.14] | [0.74,3.14] | [0.98,9.49] | | [0.49,1.36] | [0.52,2.34] | [0.39,1.25] | [0.46,2.79] |
| GIMD-10 (ref. = low deprivation) | | | | | | | | | | |
| high deprivation | | 1.42 | 1.07 | 1.35 | 1.21 | | 1.05 | 0.94 | 0.78 | 0.88 |
| | | [0.73,2.76] | [0.46,2.50] | [0.70,2.59] | [0.51,2.88] | | [0.68,1.60] | [0.52,1.70] | [0.46,1.34] | [0.41,1.89] |
| **Fixed-effects: Intercept** | **0.71\*** | | | **0.31\*\*** | 0.06 | 0.84 | | | 0.69 | 0.30 |
| | **[0.51,1.00]** | | | **[0.13,0.71]** | [0.00,1.16] | [0.68,1.03] | | | [0.36,1.33] | [0.03,3.42] |
| **Random-effects: Intercept** | 0.78 | | | 0.45 | 0.34 | . | | | . | . |
| | [0.45,1.34] | | | [0.16,1.25] | [0.03,3.61] | . | | | . | . |
| **Random-effects: Measures of variance** | | | | | | | | | | |
| Variance: accommodation | 0.61 | | | 0.20 | 0.12 | 0.00 | | | 0.00 | 0.00 |
| Variance: total | 3.90 | | | 3.49 | 3.41 | 3.29 | | | 3.29 | 3.29 |
| Intraclass correlation | 0.16 | | | 0.06 | 0.03 | 0.00 | | | 0.00 | 0.00 |
| Median Odds Ratio | 2.10 | | | 1.54 | 1.38 | 1.00 | | | 1.00 | 1.00 |

(*Continued*)

**Table 4.** (Continued)

| | Generalized Anxiety (GAD2) | | | | | Depression (PHQ2) | | | | |
|---|---|---|---|---|---|---|---|---|---|---|
| | M0 | M1a-M1e | M2a-M2e | M3 | M4 | M0 | M1a-M1e | M2a-M2e | M3 | M4 |
| | Null | Single contextual exposure, unadjusted | Single contextual exposure, adjusted | Multiple contextual exposures, unadjusted | Multiple contextual exposures, adjusted | Null | Single exposure | Single exposure, adjusted | Multi exposures, unadjusted | Multi exposures, adjusted |
| **Fixed-effects: accommodation-level** | | | | | | | | | | |
| **Model fit and sample size** | | | | | | | | | | |
| Akaike Information Criterion | 470.06 | | | 470.90 | 264.11 | 482.27 | | | 486.54 | 296.20 |
| Bayesian Information Criterion | 477.77 | | | 505.57 | 330.17 | 489.97 | | | 521.18 | 362.17 |
| Range of variance inflation factors | | | | 1.05–1.32 | 1.08–1.57 | | | | 1.03–1.30 | 1.08–1.62 |
| Wald-Chi2 | . | | | 15.05 | 34.43 | . | | | 9.34 | 16.53 |
| Model df | 0 | | | 7 | 18 | 0 | | | 7 | 18 |
| Model significance | . | | | **0.0353***| **0.0111*** | . | | | 0.2292 | 0.5558 |
| Number of observations | 348 | | | 348 | 201 | 347 | | | 347 | 200 |
| Number of clusters | 52 | | | 52 | 46 | 52 | | | 52 | 46 |

Individual-level confounder adjustments conducted for age at interview, sex, self-reported chronic illness, region of origin, educational score, number of children, asylum residence status and number of accommodation transfers.

Models M1a-M1e each report a univariate model and are reported within one column for space management. Models M2a-M2e each report on a model with one single contextual exposure variable adjusted for above mentioned individual-level confounders. Results are reported within one column. M3 reports results on one model containing all exposure variables. M4 reports results on one model containing all exposure variables and all above mentioned individual-level confounders.

95% confidence intervals in brackets. * p<0.05, ** p<0.01, *** p<0.001

SHED: Small-area Housing Environment Deterioration

GIMD-10: German Index of Multiple Deprivation, Version 10

reduction in accommodation-level variance in GAD2 after inclusion of contextual- and individual-level factors (Fig 2) towards a MOR of 1.34 in the fully adjusted model (M4) (Table 4).

## Discussion

Post-migration living conditions have been discussed as an important contextual factor for determining asylum seekers' and refugees' mental health [39]. Collective accommodation centres in particular have been considered a stress factor affecting health [11, 20, 40]. Potential causal mechanisms are discussed, ranging from overcrowding and institutional settings [41] to perceived neighbourhood disorder [42] and basic living difficulties [43].

This study utilized a quasi-random allocation of ASR into residential areas to quantitatively examine various contextual effects of housing environment on reporting symptoms of depression and generalized anxiety. We thereby minimised the risk of compositional bias through selective migration into contextual environments, to conclude on three principal findings.

## Caterpillar plot: Generalized Anxiety Disorder

**Fig 2. Plotting predicted random-intercepts of generalized anxiety disorder per accommodation centre.** Caption: Predicted random-intercepts for each accommodation centre have been ranked and plotted for three models (null model, contextually and fully adjusted models), as such visualizing the reduction in accommodation-level variance in GAD2 after inclusion of contextual- and individual-level factors.

First, housing environments for half of the participants were characterized by high deterioration, high deprivation of districts, and for most, by urbanity. About 20% of centres accommodated 50 or more residents, while 7% of centres were remote and cut-off from essential facilities (medical facilities, groceries, community town hall).

Second, about every second ASR reported symptoms of depression or anxiety, while symptoms of anxiety significantly clustered at accommodation level.

Third, we observed higher, but not statistically significant, point estimates for odds of reporting symptoms of generalized anxiety and/or depression when living in collective accommodation centres with highest level of deterioration, large numbers of residents, remote location and/or being situated in an urban district. Being in a district with high level of deprivation showed different point estimates for the two mental health outcomes: while deprivation came along with higher odds for reporting symptoms of generalized anxiety, the inverse pattern was found for symptoms of depression.

Results should be deemed exploratory and preliminary as sampling for these analyses was underpowered and fixed effects throughout lacked statistical significance. Yet, given the observed large variation in reporting symptoms of generalized anxiety on the level of accommodation centre, it remains important to conduct further research with prior sample size calculation.

While research specifically exploring contextual effects of post-migration housing environment for ASR is sparse, a number of studies have followed up on such hypotheses for the general population. For example, remoteness has been discussed as an associated factor with

depression and mental distress [44, 45]. At the same time, the exact pathways are unclear, and adverse effects on mental health may not always be through simple direct pathways.

Research constructing high-quality and finely granulated contextual housing environment data and studies utilizing advanced techniques like structural equation modelling will be important to better understand and differentiate observed and latent variables.

## Strengths and limitations

Conceived as a cross-sectional study lacking base-line data on health, causal inferences are limited, and future studies should analyse the dispersal into different contexts and potential health effects within longitudinal designs. In addition, we lacked data on length of stay of ASR in the accommodation centres which can play a critical role when assessing questions of moderation, interaction, and causality. The study was further limited by the explorative nature: point estimates of fixed effects were surrounded by large 95% CIs and lacked statistical significance, raising questions of an underpowered sampling for these analyses. Further, given the nature of the used PHQ2 and GAD2 screening instruments, all mental health symptoms were self-reported, i.e. not assessed by a mental health specialist, and may as such lack clinical relevance. While we studied the mutually adjusted effects of several contextual factors on reporting symptoms on two important mental health issues (further adjusted for individual-level factors), we did not study the interrelation or combined effects of these contextual variables on health outcomes. Given the potentially complex relationship between the explored aspects of post-migration contextual housing environment future analyses should analyse the inter-relations and potential interactions between these.

At the same time, a strength of this study lies in its high-quality and extensive data at high geographical resolution. Sampling was based on a complex design balanced for the number of ASR residing in the accommodation centre and in the respective region. The surveyed participants were representative for the overall population in the state Baden-Wuerttemberg with respect to nationality, age, and sex as reported elsewhere [2]. All instruments used are internationally established and if not were validated separately [2, 26] and, if needed, underwent a rigorous translation and pre-testing process [24]. Individual level was accurately linked to collected contextual data or to geo-referenced data at highest level of resolution, as such minimising the risk of misclassification bias. Given the resource-intense sampling procedure to reach this population, this primary data study including the linkage options offer added value to research on refugee health in the context of housing and post-migration determinants. Other studies analysing housing effects on health among ASR [20] used the IAB-BAMF-SOEP-refugee panel which has a nation-wide scope [46]. However, the majority of individuals included in the sample (> 80%) have completed asylum claims [20], which means that selective migration into housing environments in cross-sectional designs cannot be ruled out. In contrast, our sample consists of ASR who have more recently arrived in Germany; as such, only 24% of our sample had been granted residence permit and lived in the centres because they could not find or afford private housing. As freedom of movement is restricted if the decision on the asylum claim is pending, the risk of compositional bias is substantially reduced.

## Research gaps

Several research questions remain for future research studies. First, in the final exposure and confounder adjusted model M4, a median odds ratio of 1.38 remained, indicating a high unexplained accommodation-level variance. Second, given the small potentially underpowered sample for such analyses, confirmatory analyses with ex ante sample size calculations are needed. Third, to better understand the inter-relations between various post-migration

contextual housing characteristics further statistical approaches, e. g. cluster analyses, principal component analyses or other structural equation modelling techniques, may be of value. The inverse association between multiple deprivation at neighbourhood level and depression raised questions of either residual confounding or influences through other contextual factors which remained unmeasured by our study. This relates especially to higher ethnic diversity of more deprived areas, which could have a protective effect, as reported elsewhere [47, 48], through community networks and counter potential negative effects of multiple deprivation.

Finally, studies considering timing and time variance, e. g. longitudinal studies combined with randomization or utilizing natural quasi-randomization, are needed to confirm potential cause-effect relationships.

## Implications for policy

Given the adverse condition of most collective accommodation centres and pre-existing ASR's mental health burden, sustainable long-term monitoring of housing conditions and health of ASR is important to ensure a comprehensive epidemiological understanding enabling evidence-informed, efficient, and targeted preventative measures, health care spending and adequate housing policies.

Additionally, given the arrival of high numbers of Ukrainian refugees in Germany, it remains important to discuss the role of free choice of private flats and the potential health benefits entailed. In cases, in which Ukrainian refugees are staying in collective accommodation centres, it is important to discuss the situation of densely populated accommodation centres and their limited capacity to adequately respond to sharp rises in number of residents due to war and persecution.

With legal frameworks promoting compulsory assignment to districts or collective accommodation centres and living realities shaped by disempowerment and loss of autonomy, it is a moral responsibility to prevent that dispersal becomes a form of structural violence by assigning individuals into adverse and disadvantaged contexts. Housing standards and investments in maintenance of state-provided collective accommodation centres for ASR may be a critical component in ensuring long-term health maintenance and allowing post-migration contexts as "powerful determinant[s] of mental health" [8] to facilitate long-term improvements in health and building grounds for successful arrival, integration, and participation of ASR in their host country.

## Conclusion

Adjusted for individual-level factors, this exploratory study could not establish evidence for a statistically significant effect on symptoms of generalized anxiety and depression of contextual factors such as higher levels of housing deterioration, larger numbers of residents, remoteness of accommodation centres and district-level of urbanity. High deprivation at district-level may be a risk factor for generalized anxiety but potentially a protective factor for reporting symptoms of depression.

Residual confounding by length of stay in the accommodation centre cannot be ruled out in our analyses. Unmeasured variation in length of stay in the centres is likely to conflate possible relationships between exposure (e. g. housing deterioration) and outcome as some time may be required from being exposed and symptoms to be developed. New arrivals and shorter stays are hence likely to mask the effects on those who lived in the same centre for a longer time.

Together, the examined factors reduced variance at accommodation level, but further unexplained variance remained for reporting symptoms of generalized anxiety. Further confirmatory studies with ex ante sample size calculations and longitudinal designs are needed.

## Author Contributions

**Conceptualization:** Amir Mohsenpour, Louise Biddle, Kayvan Bozorgmehr.

**Data curation:** Amir Mohsenpour, Louise Biddle.

**Formal analysis:** Amir Mohsenpour.

**Funding acquisition:** Kayvan Bozorgmehr.

**Methodology:** Amir Mohsenpour, Louise Biddle, Kayvan Bozorgmehr.

**Project administration:** Louise Biddle, Kayvan Bozorgmehr.

**Supervision:** Kayvan Bozorgmehr.

**Writing – original draft:** Amir Mohsenpour.

**Writing – review & editing:** Louise Biddle, Kayvan Bozorgmehr.

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
