## [Decision Letter · Decision Letter 0]

28 Dec 2022

PGPH-D-22-01853

Exploring contextual effects of post-migration housing environment on mental health of asylum seekers and refugees: a cross-sectional, population-based, multi-level analysis in a German federal state

Dear Dr. Amir Mohsenpour,

Thank you for submitting your manuscript to PLOS Global Public Health. After careful consideration, we feel that it has merit but does not fully meet PLOS Global Public Health’s publication criteria as it currently stands. Therefore, we invite you to submit a revised version of the manuscript that addresses the points raised during the review process.

We look forward to receiving your revised manuscript.

Kind regards,

Oliver Mendoza-Cano, Sc. D.

Academic Editor

Journal Requirements:

a. Please clarify all sources of funding (financial or material support) for your study. List the grants (with grant number) or organizations (with url) that supported your study, including funding received from your institution. 

b. State the initials, alongside each funding source, of each author to receive each grant.

c. State what role the funders took in the study. If the funders had no role in your study, please state: “The funders had no role in study design, data collection and analysis, decision to publish, or preparation of the manuscript.”

d. If any authors received a salary from any of your funders, please state which authors and which funders.

2. Since your data is not available for proprietary reasons, please explain via email why the data is not available. Please also include the contact information for the third party organization that should be contacted should other researchers want to request access to this data and please include the full citation of where the data can be found. We also request that you verify with us via email that any researcher will be able to obtain the data set in the same manner that the you have obtained it. If you feel you are unwilling or unable to adhere to this policy, please explain your reasons by return email and your exemption request will be escalated to the editor for approval. Your exemption request will be handled independently and will not hold up the peer review process, but will need to be resolved should your manuscript be accepted for publication. One of the Editorial team will be in touch if they require more information.

Additional Editor Comments (if provided):

Reviewer 1

The evaluated manuscript aimed to analyze contextual effects of post-migration housing environment on their mental health.

Congratulations to the authors because I consider that it is an interesting manuscript and of great public health relevance, however I must make the following observations and suggestions:

1. In the abstract, the authors list the results of the ORs that the model yielded, however none of these show a statistically significant result. I consider that it is useless to place these values and it would be more appropriate to describe the results of the variables corresponding to the second level of analysis and finally mentioning that none had effects on the mental health of the population analyzed in this study.

2. What does it mean for the authors that a tendencies were found? I suggest avoiding this phrase in the presence of statistical results without significance. It is prudent to point out that the frequencies and proportions are higher in a group with depression or anxiety, but the word “tendencies” is ambiguous.

3. The methods are appropriately described and the use of multilevel analysis is adequately presented, some doubts about the variables that characterize each level arise when reading them are

a. Why was a sample size not calculated?

b. Why were the centers dichotomized into ≥50 people per center? Could you explain what is the usual population that exists in these centers on average?

c. The region of origin definition that is analyzed is very broad and perhaps does not reflect the factor that could really be influencing, for example, coming from a low-income country or at war.

d. Why was 20 minutes considered as the time to calculate the remoteness index? What is the optimal time for these transfers in Germany?

4. Regarding the results, in Table 3 "Sample description", where the frequencies and proportions for the variables of each level are described, the number of participants do not coincide with the total sample size shown in Table 2

5. The titles of the tables do not adequately inform their content

6. Figures must have their own caption containing the explanation found in the text.

7. The discussion of this work is very limited and I consider it its greatest weakness, the authors do not compare their results with those found in other populations, they do not consider that in reality the post-migration housing environment has no effect on depression or anxiety and what would be the possible explanations for this in this population of Germany.

8.The conclusion must be rewritten because it is ambiguous and the results of the analysis are clear.

Reviewer 2

The authors abide to the truth and are honest about the absence of significant differences, which should be praised. However, and due to its limited findings, this manuscript should be submitted only as a short communication, which could be useful to other scientists in the designing of similar studies.

Reviewer 3

This original article explored the effect of post-migration on anxiety and depression symptoms (and other sociodemographic aspects) in asylum seekers and refugees in a German federal state. The proposed results show a clear need for improvement of well-being of this population. Some recommendations are presented below.

-If possible, the symptoms of anxiety and depression (GAD-2, PHQ-2) could be further explored if descriptive data are presented on each item analyzed in the study participants .

-It is required not to generalize to "mental health" only with the presence of anxiety and depression. In line 120 it could be clarified in a more detailed way.

-I recommend adding a univariate analysis of Table 2, to identify the associations between the presented variables .

-The discussion section requires further analysis of all the variables analyzed. It is required that the results obtained be hypothesized and compared with those of other authors.

-I recommend reducing the extension of the analysis of the generalized linear models, since most of the results did not have statistical significance and/or the confidence intervals presented do not have enough statistical power to take them as valid.

-Place the corresponding abbreviations at the foot of tables 2 and 3.

---

## [Decision Letter · Decision Letter 1]

24 May 2023

PGPH-D-22-01853R1

Exploring contextual effects of post-migration housing environment on mental health of asylum seekers and refugees: a cross-sectional, population-based, multi-level analysis in a German federal state

Dear Dr. Mohsenpour,

Thank you for submitting your manuscript to PLOS Global Public Health. After careful consideration, we feel that it has merit but does not fully meet PLOS Global Public Health’s publication criteria as it currently stands. Therefore, we invite you to submit a revised version of the manuscript that addresses the points raised during the review process.

Your manuscript has been evaluated by three reviewers, including two of the previous reviewers plus one new reviewer; their comments are appended below.

While all reviewers are overall positive towards publication, Reviewer #3 has provided comments recommending further revisions to clarify some matters in the Methods and Discussion sections. You may also want to consider responding to the comment from Reviewer #2 regarding future studies into the generalizability of the findings in this study.

We look forward to receiving your revised manuscript.

Kind regards,

Hugh Cowley

Staff Editor

Journal Requirements:

Additional Editor Comments (if provided):

Reviewers' comments:

Reviewer's Responses to Questions

**Comments to the Author**

1. If the authors have adequately addressed your comments raised in a previous round of review and you feel that this manuscript is now acceptable for publication, you may indicate that here to bypass the “Comments to the Author” section, enter your conflict of interest statement in the “Confidential to Editor” section, and submit your "Accept" recommendation.

Reviewer #1: All comments have been addressed

Reviewer #2: All comments have been addressed

Reviewer #4: (No Response)

2. Does this manuscript meet PLOS Global Public Health’s publication criteria? Is the manuscript technically sound, and do the data support the conclusions? The manuscript must describe methodologically and ethically rigorous research with conclusions that are appropriately drawn based on the data presented.

Reviewer #1: Yes

Reviewer #2: Yes

Reviewer #4: Yes

3. Has the statistical analysis been performed appropriately and rigorously?

Reviewer #1: Yes

Reviewer #2: Yes

Reviewer #4: Yes

4. Have the authors made all data underlying the findings in their manuscript fully available (please refer to the Data Availability Statement at the start of the manuscript PDF file)?

Reviewer #1: Yes

Reviewer #2: Yes

Reviewer #4: No

5. Is the manuscript presented in an intelligible fashion and written in standard English?

Reviewer #1: Yes

Reviewer #2: Yes

Reviewer #4: Yes

6. Review Comments to the Author

Reviewer #1: The authors have adequately responded to the comments, I consider that now the manuscript can be published in the present form

Reviewer #2: I find very intriguing the finding about a "protective" effect of deprivation before depression; however, despite the study recognizes its limitations, its findings warrant future studies to determine if the tendencies found are consistent among ASR, not only in Germany, but in other countries.

Reviewer #4: Dear Authors,

Thanks for sharing this manuscript. I enjoyed reading it.

Your manuscript provides an important overview of the effects of the post-migration housing environment on the mental health of asylum seekers and refugees in Germany. Housing situation is a determinant of health that is underreported in the context of migrant health. The study has many merits; however, I recommend the authors address the following points in their manuscript:

Methods:

- The authors referred to previously reported details about "Setting, sampling and recruitment". However, they should provide more details about their study's setting and data collection. This should include the data collection period, the languages into which the questionnaire was translated, and the rationale for choosing these languages.

- Did the authors involve the concerned population (asylum seekers and refugees) in the design and implementation of their study? If not, why? This should be reflected in the limitations.

- The length of stay in the collective accommodation centres is a potential confounder that has yet to be considered in the analyses. Authors should explain why they did not include this confounder and reflect on this in the limitations.

Discussion:

- How would this study's outcomes apply to other parts of Germany?

- Could the authors put their study in the context of the current situation of asylum seekers and refugees in Germany, considering the high number of people who fled Ukraine to Germany?

Best wishes for you resubmission.

7. PLOS authors have the option to publish the peer review history of their article (what does this mean?). If published, this will include your full peer review and any attached files.

**Do you want your identity to be public for this peer review?** For information about this choice, including consent withdrawal, please see our Privacy Policy.

Reviewer #1: No

Reviewer #2: No

Reviewer #4: No

---

## [Decision Letter · Decision Letter 2]

21 Aug 2023

PGPH-D-22-01853R2

Exploring contextual effects of post-migration housing environment on mental health of asylum seekers and refugees: a cross-sectional, population-based, multi-level analysis in a German federal state

Dear Dr. Mohsenpour,

Thank you for submitting your manuscript to PLOS Global Public Health. After careful consideration, we feel that it has merit but does not fully meet PLOS Global Public Health’s publication criteria as it currently stands. Therefore, we invite you to submit a revised version of the manuscript that addresses the points raised during the review process.

We look forward to receiving your revised manuscript.

Kind regards,

Jianhong Zhou

Staff Editor

Journal Requirements:

Additional Editor Comments (if provided):

Reviewers' comments:

Reviewer's Responses to Questions

**Comments to the Author**

1. If the authors have adequately addressed your comments raised in a previous round of review and you feel that this manuscript is now acceptable for publication, you may indicate that here to bypass the “Comments to the Author” section, enter your conflict of interest statement in the “Confidential to Editor” section, and submit your "Accept" recommendation.

Reviewer #1: All comments have been addressed

Reviewer #2: All comments have been addressed

Reviewer #4: (No Response)

2. Does this manuscript meet PLOS Global Public Health’s publication criteria? Is the manuscript technically sound, and do the data support the conclusions? The manuscript must describe methodologically and ethically rigorous research with conclusions that are appropriately drawn based on the data presented.

Reviewer #1: Yes

Reviewer #2: Yes

Reviewer #4: (No Response)

3. Has the statistical analysis been performed appropriately and rigorously?

Reviewer #1: Yes

Reviewer #2: Yes

Reviewer #4: (No Response)

4. Have the authors made all data underlying the findings in their manuscript fully available (please refer to the Data Availability Statement at the start of the manuscript PDF file)?

Reviewer #1: Yes

Reviewer #2: Yes

Reviewer #4: (No Response)

5. Is the manuscript presented in an intelligible fashion and written in standard English?

Reviewer #1: Yes

Reviewer #2: Yes

Reviewer #4: (No Response)

6. Review Comments to the Author

Reviewer #1: All comments have been addressed

Reviewer #2: (No Response)

Reviewer #4: Dear Authors,

Thanks for sharing your revised manuscript. It is great to see how you have addressed many of the questions I shared. There are still some concerns to be further addressed in your manuscript.

Best wishes for your resubmission.

- The length of stay in the collective accommodation centres is a potential confounder that

has yet to be considered in the analyses. Authors should explain why they did not include this

confounder and reflect on this in the limitations.

We have added details on the lack of data on length of stay in the limitations section

(lines323-324).

Great that you have added this statement to refer to this limitation of your study. This is not a minor limitation. Please refer to this limitation in your abstract and conclusion and try to reflect on how you think the following outcome of your study: “No significant clustering was found for symptoms of depression” would be influenced by including the length of stay in collective accommodation centres.

- Could the authors put their study in the context of the current situation of asylum seekers

and refugees in Germany, considering the high number of people who fled Ukraine to

Germany?

A direct comparison or contrasting of the different ASR populations is out of scope for this

manuscript. The current Ukrainian refugees arrive(d) in Germany with a protected status as

war refugees and do not need to apply for asylum in Germany. As such they are not part of

the usual trajectory in terms of housing, are not placed in the initial state-wide reception

centres, and have high residential mobility.

I am not suggesting comparing your study population with other populations. Some Ukrainian refugees are staying in some of the collective accommodation centres where you collected the data for your study. Any reflection on how did the arrival of high numbers of Ukrainian refugees to Germany impact the living conditions in collective accommodation centres?

Best wishes for you resubmission.

7. PLOS authors have the option to publish the peer review history of their article (what does this mean?). If published, this will include your full peer review and any attached files.

**Do you want your identity to be public for this peer review?** For information about this choice, including consent withdrawal, please see our Privacy Policy.

Reviewer #1: No

Reviewer #2: No

Reviewer #4: No

---

## [Decision Letter · Decision Letter 3]

27 Nov 2023

Exploring contextual effects of post-migration housing environment on mental health of asylum seekers and refugees: a cross-sectional, population-based, multi-level analysis in a German federal state

PGPH-D-22-01853R3

Dear Mr. Mohsenpour,

We are pleased to inform you that your manuscript 'Exploring contextual effects of post-migration housing environment on mental health of asylum seekers and refugees: a cross-sectional, population-based, multi-level analysis in a German federal state' has been provisionally accepted for publication in PLOS Global Public Health.

Best regards,

Julia Robinson

Executive Editor

Reviewer Comments (if any, and for reference):

Reviewer's Responses to Questions

**Comments to the Author**

1. If the authors have adequately addressed your comments raised in a previous round of review and you feel that this manuscript is now acceptable for publication, you may indicate that here to bypass the “Comments to the Author” section, enter your conflict of interest statement in the “Confidential to Editor” section, and submit your "Accept" recommendation.

Reviewer #1: All comments have been addressed

Reviewer #2: All comments have been addressed

Reviewer #4: All comments have been addressed

2. Does this manuscript meet PLOS Global Public Health’s publication criteria? Is the manuscript technically sound, and do the data support the conclusions? The manuscript must describe methodologically and ethically rigorous research with conclusions that are appropriately drawn based on the data presented.

Reviewer #1: Yes

Reviewer #2: Yes

Reviewer #4: Yes

3. Has the statistical analysis been performed appropriately and rigorously?

Reviewer #1: Yes

Reviewer #2: Yes

Reviewer #4: Yes

4. Have the authors made all data underlying the findings in their manuscript fully available (please refer to the Data Availability Statement at the start of the manuscript PDF file)?

Reviewer #1: Yes

Reviewer #2: Yes

Reviewer #4: (No Response)

5. Is the manuscript presented in an intelligible fashion and written in standard English?

Reviewer #1: Yes

Reviewer #2: Yes

Reviewer #4: Yes

6. Review Comments to the Author

Reviewer #1: My observations were addressed from previous versions, and I consider that residual observations made by another reviewer were addressed now

Reviewer #2: All corrections have been addressed. I recommend the manuscript to be published without further corrections.

Reviewer #4: Dear Authors,

Thanks for adequately addressing the comments I shared.

7. PLOS authors have the option to publish the peer review history of their article (what does this mean?). If published, this will include your full peer review and any attached files.

**Do you want your identity to be public for this peer review?** For information about this choice, including consent withdrawal, please see our Privacy Policy.

Reviewer #1: No

Reviewer #2: No

Reviewer #4: No
